# Self-Powered Gradient Hydrogel Sensor with the Temperature-Triggered Reversible Adhension

**DOI:** 10.3390/polym14235312

**Published:** 2022-12-05

**Authors:** Dong Sun, Cun Peng, Yuan Tang, Pengfei Qi, Wenxin Fan, Qiang Xu, Kunyan Sui

**Affiliations:** 1State Key Laboratory of Bio-Fiber and Eco-Textiles, Collaborative Innovation Center for Marine Biobased Fibers and Ecological Textile Technology Institute of Marine Biobased Materials, College of Materials Science and Engineering, Qingdao University, Qingdao 266071, China; 2College of Chemistry and Bioengineering, Hunan University of Science and Engineering, Yongzhou 425199, China; 3Department of Ophthalmology, The Affiliated Hospital of Qingdao University, Qingdao 266071, China

**Keywords:** artificial skin, self-powered gradient hydrogels, temperature-triggered adhesion, mechanical sensitive sensing

## Abstract

The skin, as the largest organ of human body, can use ions as information carriers to convert multiple external stimuli into biological potential signals. So far, artificial skin that can imitate the functionality of human skin has been extensively investigated. However, the demand for additional power, non-reusability and serious damage to the skin greatly limits applications. Here, we have developed a self-powered gradient hydrogel which has high temperature-triggered adhesion and room temperature-triggered easy separation characteristics. The self-powered gradient hydrogels are polymerized using 2-(dimethylamino) ethyl metharcylate (DMAEMA) and N-isopropylacrylamide (NIPAM) under unilateral UV irradiation. The prepared hydrogels achieve good adhesion at high temperature and detachment at a low temperature. In addition, according to the thickness-dependent potential of the gradient hydrogel, the hydrogels can also sense pressure changes. This strategy can inspire the design and manufacture of self-powered gradient hydrogel sensors, contributing to the development of complex intelligent artificial skin sensing systems in the future.

## 1. Introduction

As an important organ of the human body, the skin can not only protect the human body from external bacteria and viruses, but also accurately receive and sense various external physical and chemical stimuli. To date, people have developed multiple artificial skins to realize the function of human skin [1,2,3]. Among others, the self-powered sensors based on piezoelectric [4,5,6,7,8,9] and triboelectric nanogenerators (TENG) [10,11,12,13] have attracted great attention due to their miniaturization, high sensitivity, simple preparation and lack of an external power supply. However, these traditional self-powered sensors based on stretchable piezoelectric and triboelectric nanogenerators transmit signals electronically, which is different from the signal carrier (i.e., ions) of the human sensory system. We previously constructed self-powered ionic sensors based on gradient polyelectrolyte hydrogels [14,15]; however, beyond that, it is also urgent to build a revisable adhension self-powered hydrogel ionic sensor when applied to human body.

Traditional artificial skin adhesives mainly include bionic microstructure adhesives inspired by octopuses [16,17,18], tree frogs [19] and other organisms [20,21,22], pressure-sensitive adhesives [23], and chemical adhesives [24,25,26]. However, these traditional adhesives are commonly disposable equipment. They have poor adhesion on soft surfaces, and may be difficult to remove after use. In the process of stripping, they may cause pain and serious inflammation and cannot be reused after removal. It has been reported that these separable adhesive materials constituted of non-covalent or dynamic covalent bonds can break or recombine under the action of external stimuli such as specific solutions [27], ultraviolet rays [28,29], electric currents [30] and magnetic fields [31]. Although these artificial skins have reversible adhesive properties, they need to use external stimuli, which may not be available to the human body, or the conditions are harsh and are limited to specific types of substrates. Therefore, thermal-responsive hydrogels with adjustable temperature trigger mechanisms and signal recognition abilities have come into vision [32,33]. For example, Wu et al. [34] designed a thermal-responsive hydrogel of an upper critical solution temperature (UCST) type. The hydrogels were constructed by in situ polymerization of an acrylic monomer in quaternized chitosan solution. The thermal response behavior and adjustable adhesion ability of the hydrogels were achieved by changing the hydrogen bond interaction in the reversible phase transition process. However, there are few reports on thermal responsive hydrogel sensors that combine self-powered structure and adjustable adhesion capability.

Here, we have developed self-powered gradient hydrogel sensors with a reversible adhension capacity based on the copolymers with phase transition temperature, which are polymerized by 2-(dimethylamino)ethyl metharcylate (DMAEMA) and N-isopropylacrylamide (NIPAM). The hydrogels have great adhesion through their own hydrophobicity, where the amino group of the hydrogel forms a hydrogen bond with the substrates, and the ammonium ion forms an electrostatic complexation when the human body’s temperature is higher than the lower critical solution temperature (LCST). The hydrogel adhesive has high viscosity above 37 °C, can maintain the adhesive state, and has reduced adhesion at room temperature, which means that it is easy to remove from the skin. This study demonstrates the potential application of the developed hydrogel adhesives in artificial skin and self-powered biomechanical monitoring system.

## 2. Experimental Section

### 2.1. Materials

2-(Dimethylamino)ethyl metharcylate (DMAEMA, 99%, remove MEHQ stabilizer) and 2-hydroxy-4′-(2-hydroxyethoxy)-2-methylpropiophenone (2959, ≥98.0%, HPLC) were purchased from Aladdin Reagents Co. (Shanghai, China). N-Isopropylacrylamide (NIPAM, stabilized with MEHQ) was purchased from TCI (Shanghai, China). *N*,*N*-Methylenebisacrylamide (MBAA, 99%) and the UV absorber (UV F-22) were purchased from Weihai Huaen Rubber Plastic New Material Co. (Weihai, China) Alkaline alumina was used (200–500 mesh).

### 2.2. Preparation of the Gradient Polyelectrolyte Hydrogels

The gradient PDMAEMA/PNIPAM hydrogels were fabricated by free radical polymerization under UV irradiation for 4 h and in the presence of a UV adsorber. In brief, the total content of the two monomers in the solution was 30 wt %, and the ratio of DMAEMA to NIPAM was 10:1, such that there was 0.15 wt % MBAA (cross-linker), 0.3 wt % 2959 (photoinitiator) and 0.1 wt % UV absorbers. These were sequentially dissolved into deionized water and the mixed solution was stirred. After standing, the precursor solution was injected into a home-made mold consisting of two glasses and a spacer. Then, then the mold was placed under 360 nm ultraviolet light for 4 h to polymerize. The PDMAEMA/PNIPAM gradient hydrogel was finally obtained by radical polymerization under ultraviolet light.

### 2.3. Preparation of Self-Powered Sensors

The self-powered hydrogel sensor can be easily integrated by connecting the upper and lower sides of the hydrogel to two copper foil electrodes respectively. During the sensing processes, the hydrogel sensors were attached to the tested solid or skin surface by commercial biaxially oriented polypropylene (BOPP) tape and copper foil tape. The stress and strain sensing performance as well as the large deformation of human body based on the hydrogel sensors were measured.

### 2.4. Characterization

The cross-sectional morphologies of the PDMAEMA/PNIPAM gradient hydrogels were observed using a scanning electron microscope (SEM, Quanta 250 FEG). The element content distribution on the upper and lower surfaces of the gradient hydrogels were analyzed using X-ray photoelectron spectroscopy (XPS, Thermo Scientific Escalab 250Xi+, Waltham, MA, USA). The transparency of the receptors was characterized by an ultraviolet and visible spectrophotometer (UV-vis, T9). The mechanical properties were characterized with the strip-like hydrogel using a testing machine (WDW-5T) at tensile speed of 5 mm/min. The peel strength of the hydrogel film was measured with a peeling speed of 20 mm/min using a mechanical testing machine (WDW-5T). The electrochemical measurements were performed on a CHI660 electrochemical workstation (CH Instruments, Inc. Shanghai, China). The sensing performance and the output voltage of the receptors regarding external stimulus such as pressure and strain were monitored by a digital source meter (Keithley 2450, Shanghai, China).

## 3. Results and Discussion

### 3.1. Preparation and Characterization 

The hydrogels were prepared by free radical polymerization, and the gradient structure was induced by ultraviolet light irradiation in the presence of an absorber (Figure 1). Firstly, the monomer of DMAEMA was disposed through alkaline alumina to remove the stabilizer. The homogeneous precursor solution composed of DMAEMA, NIPAM, 2959, MBAA, and the UV absorber was prepared and injected into the home-made molds consisting of two transparent glass plates and a spacer. The polymerization reaction was irradiated by single-side UV irradiation for 4 h, and the monomers of DMAEMA and NIPAM were cross-linked by covalent bonds. Meanwhile, the strong ultraviolet absorption of the absorber will cause the light intensity to attenuate along the thickness/irradiation direction, leading to the as-prepared hydrogels with gradient structure. In other words, the UV irradiation side (denoted as HD side) of the hydrogel has a higher polymerization rate and a denser network than the non-ultraviolet radiation side (denoted as the LD side). If the UV irradiation time is too short, the polymerization time will be short and the hydrogels cannot be obtained. If the UV irradiation time is long enough, and the gradient structure of the hydrogel will not be clear. Additionally, the mechanical properties of the hydrogel will become worse (as shown in Appendix A), and the tensile strength will become lower, which will not reach the mechanical properties we imagined. Thus, the gradient structure of the hydrogels can be regulated via the content of the UV adsorber and illumination time.

The gradient structure of the hydrogels was characterized by SEM and XPS, respectively. The HD side of the hydrogel has a denser polymer network than the other side, according to the cross-sectional SEM image of the hydrogels (Figure 2a). In addition, the element contents on both sides of the hydrogel were examined and compared through XPS analysis. Compared with the LD side, the HD side of the gradient PDMAEMA/PNIPAM hydrogel shows a greater carbon content and lower nitrogen content and oxygen content, which further proves that the higher chain density of PDMAEMA on this side (Figure 2b). These results suggest the we have successfully prepared the hydrogel with gradient structure along with the thickness direction.

It is important to manipulate the phase transition temperature of the hydrogel to be close to the temperature of human body. The transition temperature of the hydrogel was tuned by modifying the DMAEAM:NIPAM weight ratio and was measured via turbidity measurements. Figure 2c shows the transmittance of the as-prepared hydrogels vs. temperature variation. PNIPAM is a typical thermosensitive monomer, and the LCST of the hydrogel prepared by pure PNIPAM through turbidity measurement is about 31 °C [35,36]. While when the mass fraction of NIPAM decreased from 5:1 to 15:1, the LCST of the hydrogels increased from 30 °C to 34 °C (Figure 2c). The unique phase transition characteristics of hydrogel enable it to be treated as an independent membrane at room temperature, and it has high transparency so that the words and patterns behind it can be clearly seen. When the temperature of the hydrogel is increased beyond its LCST, its transparency decreases significantly (Figure 2d). Therefore, we have successfully prepared gradient hydrogels with adjustable LCST that can be treated as membranes at room temperature.

The mechanical properties of gradient hydrogel membranes were further investigated to ensure that they can be used at a higher temperature than LCST. Their tensile mechanical properties at different temperatures were measured, respectively. As shown in Figure 2e,f, with the increase of DMAEMA concentration from 5:1 to 15:1, the tensile strength and elastic modulus of the hydrogel was significantly improved, and the tensile strength and elastic modulus of the hydrogel was further enhanced when the temperature rose to 37 °C. This is because the gradient hydrogel loses water and the molecular chain tangles at a higher temperature, which can dissipate energy and disperse stress. Then, we carried out 100 stress–strain cyclic tensile tests on gradient hydrogels with different concentrations, and this showed that the hydrogel can return to its original shape after the stress is removed after stretching to 200% strain (Figure 2g–i). According to its mechanical properties and its LCST, the hydrogels prepared with monomer ratio of 10:1 were chosen for the subsequent experiments. In conclusion, the gradient hydrogels have good mechanical properties both at room temperature and high temperature, and can adapt to the movement and deformation of skin and joints.

### 3.2. Adhesion Properties of the Hydrogels

The gradient PDMAEAM/PNIPAM hydrogels can realize the reversible adhesive on various solid surfaces by adjusting the temperature due to their thermal sensitivity. The hydrogel films are adhered to flat or bent insulating substrates, including platinum, copper, silicone rubber, and glasses by sticking and gently pressing (Figure 3a–d). In order to quantitatively evaluate the adhesive strength of the hydrogel, we conducted the peel test at a peel rate of 20 mm/min. The peel strength is calculated by dividing the peel force (F) by the film width (w) [37,38]. In a typical experiment, the tested hydrogel and the base material are respectively fixed in two clamps, and the other end of the 1 mm thick hydrogel is adhered to the base material. The measured bonding strength between the hydrogels and substrates are shown in Figure 3(e1–h1). It can be seen that when the heating temperature rises to 37 °C, the average peel strength between the hydrogels and platinum, copper, silica gel, and glass is far greater than the average peel strength at room temperature. Figure 3(e2–h2) shows that the maximum interfacial adhesion strength between the hydrogels and the substrates of platinum, copper, silicone rubber and glass at 37 °C can reach 15.7, 15.5, 18.8, and 317.8 J/m^2^ respectively, which is much higher than that at room temperature. This is because the hydrogel loses water at high temperature, and the hydrogen bonds and electrostatic complexation of the hydrogel increase. With the increase of the proportion of DMAEMA monomer, the interaction of hydrogen bonds increase, the average peel strength between the hydrogel and various substrate materials is also increasing. When standing in ambient air for a while or cooling with water to room temperature, the hydrogen bonds between the hydrogel and the base material break, the average peel strength decreases, and the hydrogel is easy to peel. Therefore, no matter the cooling method, the adhesion between the hydrogel at room temperature and the tested substrate surface is poor. Therefore, in addition to its application in skin devices, the PDMAEMA/PNIPAM hydrogels can also be used in other specific scenes such as the various substrates requiring adjustable adhesion. To illustrate the reversible adhesion capacity of gradient hydrogel, we used DMAEMA: NIPAM = 10:1 hydrogel with silicone rubber as the adhesive substrate and carried out 10 repeated peel tests on it. It was found that the interfacial toughness fluctuates slightly and only decreases slightly, as shown in Appendix A. This shows that it has good adhesion repeatability.

### 3.3. Sensing Mechanism of Self-Powered Gradient Hydrogel

It is important to explore the sensing mechanism of the self-powered gradient hydrogels. The self-powered potential of PDMAEMA/PNIPAM gradient hydrogel can be calculated as follows:(1)$$\Delta \varphi =\frac{RT}{F}ln\frac{c_{Ac-}\left(h\right)}{c_{Ac-}\left(l\right)}$$
where *F*, *R* and *T* denote the Faraday constant, gas constant, and temperature respectively, and *C_Ac-_*(*h*) and *C_Ac-_*(*l*) are ions (OH^−^) concentration on HD side and LD side of gradient hydrogel respectively.

The gradient hydrogels can be considered as an aggregate of many ultrathin homogeneous layers, whose density of charged groups increases gradually along the vertical direction (Figure 4). According to Formula (1), the potential difference between any two adjacent ultra-thin layers (*k*^th^ and (*k* + 1)^th^) can be expressed as:(2)$$\Delta \varphi =\frac{RT}{F}ln\frac{c_{Ac-}\left(k+1\right)}{c_{Ac-}\left(k\right)}$$
where *C_Ac-_(k)* and *C_Ac-_*(*k* + 1) represent the concentrations of OH^−^ at the middle of *k*^th^ layer and (*k + 1*)^th^ layer. Therefore, the total built-in potential of gradient hydrogel can be converted into
(3)$$\Delta \varphi =\overset{n-1}{\underset{k=1}{\sum }}\Delta \varphi \left(k\right)=\frac{RT}{F}ln\frac{c_{Ac-}\left(n\right)}{c_{Ac-}\left(1\right)}$$

In the formula, *C_Ac-_*(*n*) and *C_Ac_*_-_(1) represent OH^−^ concentration on the upper surface of the n-th ultrathin layer and the bottom surface of the first ultrathin layer respectively. When the sensor is stretched or squeezed, the thickness of each ultrathin layer decreases, leading to a reduction in diffusion distance (Figure 4), and more OH^−^ diffuses from the (*k* + 1)^th^ layer to the *k*^th^ layer (or from *n*^th^ layer to 1th layer). Therefore, the ratio of *C_Ac-_*(*k* + 1)/*C_Ac-_*(*k*) and *C_Ac-_*(*n*)/*C_Ac_*_-_(1) is decreased, resulting in a decrease in the output voltage of the sensor when it is stretched or pressed.

### 3.4. Pressure- and Strain-Sensing Performance

The gradient PDMAEMA/PNIPAM hydrogels also can be applied as self-powered ionic sensors as their self-induced potential varies with thickness. When an external pressure is applied to the self-powered hydrogel sensors, its maximum output voltage (i.e., open circuit voltage) gradually decreases along with the reduction of thickness (Figure 5a). The pressure sensitivity can be defined as SP = |ΔV/ΔP|, where ΔV and ΔP denote the change of output voltage and applied pressure, respectively. The pressure sensitivity of the hydrogel sensor is 106.46 mV/MPa when the pressure is lower than 0.19 MPa, and it drops to 5.09 mV/MPa when the pressure exceeds 0.63 MPa. Different from the traditional electronic skin sensors based on triboelectricity or piezoelectricity, the self-powered hydrogel sensor based on PDMAEMA/PNIPAM shows excellent ability in detecting static or low-frequency dynamic mechanical stimuli. When a constant pressure is applied to the receiver, its output voltage will decrease to a stable value with slight fluctuations. After removing the pressure, the output voltage returns to the initial value (Figure 5b). The response time and recovery time of the hydrogel sensor were 1.20 s and 1.21 s, respectively, which indicates that it can detect low-frequency/static mechanical stimulation. They also have a good reversible response capability. They can respond to a very small pressure of 0.2 kPa and generate repeatable waveforms under repeated loading/unloading processes in a wide pressure range (0.2–8 kPa) (Figure 5c). In addition, this self-powered hydrogel has excellent durability and stability. The self-powered hydrogel based on PDMAEMA/PNIPAM can still maintain stable electrical response for more than 200 cycles under repeated loading and unloading pressure of 5 kPa (Figure 5d). 

The self-powered hydrogel sensors can also be used to detect strain changes. Its maximum output voltage decreases during stretching (Figure 5e). Strain sensitivity (S_ε_) is defined as S_ε_ = |ΔV/Δε|, where Δε is the change of the applied strain. A relatively high strain sensitivity of 2.71 can be obtained under a small strain of less than 40%. When the strain increases to 100%, the strain sensitivity decreases to 1.53. The hydrogel sensor has a wide strain sensing range and can generate repeatable waveforms to respond to 10% to 100% strain (Figure 5f). In addition, the self-powered hydrogel sensor based on PDMAEAM/PNIPAM also has a long current duration, with a full width at half peak (FWHM) of 5.8 s, while the FWHM of the triboelectric and piezoelectric sensors is about 100 ms [39,40] (Figure 5g). We performed an electrochemical impedance spectroscopy (EIS) test on the self-powered hydrogel. Appendix A presents Nyquist plots of the ionic films with different DMAEMA:NIPAM mass ratios measured by the EIS, where the impedance of hydrogels obtained from the intercepts of EIS curves with the *x*-axis decreased with increasing NIPAM concentration and the corresponding ionic conductivity also rose. We also compared the prepared self-powered hydrogel sensor with a piezoelectric sensor and a triboelectric sensor. Our self-powered hydrogel can achieve temperature triggered reversible adhesion while ensuring a better stress strain sensitivity and sensing range (Appendix A).

Based on the excellent pressure and strain sensing performance, these hydrogel sensors can be used as a self-powered ionic skin to monitor various human movements. Because of the wide range of strain detection, they can track the bending movement of fingers, wrists, and other joints in real time regardless of the bending speed (Figure 5h,i). For example, when installed on the finger, the output voltage can be observed to decrease gradually with the gradual increase of the finger bending angle (Figure 5h). At the same time, the hydrogel sensor can also accurately record the flexible movement of the wrist joint (Figure 5i).

## 4. Conclusions

In conclusion, a self-powered gradient hydrogel with good reversible adhesion was prepared via free radical polymerization and was induced under unilateral ultraviolet irradiation. When the environmental temperature is higher than the LCST (34 °C), the amino groups of DMAEMA can form hydrogen bonds with the substrate materials, and the ammonium ions form electrostatic complexation. The hydrophobicity also contributes to the adhesion ability of the hydrogels. When the temperature drops below the LCST, the hydrogel becomes easy to separate due to the breaking of hydrogen bonds. Additionally, benefiting from the thickness-dependent potential of gradient hydrogel, the sensors can precisely perceive a tiny variation in pressure and accurately sense tiny changes in strain. The hydrogel sensors can sense the changes of human fingers, wrists, and other joints. This study proves the potential application of self-powered gradient hydrogel in artificial skin and human biological monitoring system.

## Figures and Tables

**Figure 1 polymers-14-05312-f001:**
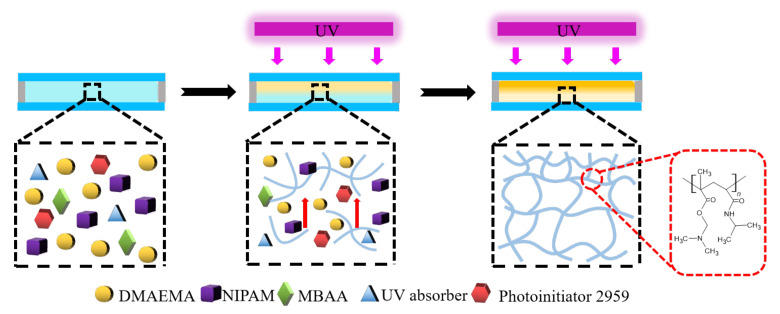
Preparation of self-powered gradient hydrogels with a reversible adhesion capacity.

**Figure 2 polymers-14-05312-f002:**
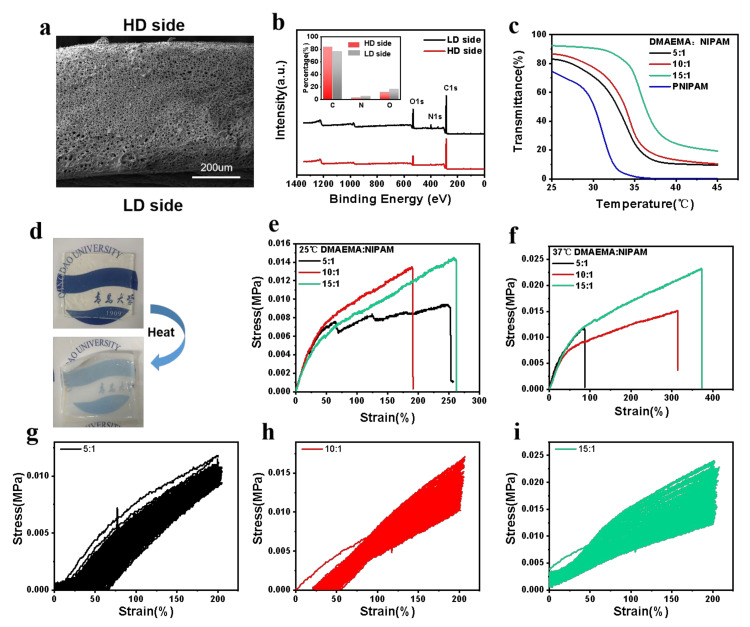
Characterization of the self-powered gradient hydrogel. (**a**) Cross-sectional SEM images of the gradient PDMAEMA/PNIPAM hydrogel. (**b**) Wide-scan survey XPS spectra of the HD side and LD side for the gradient PDMAEMA/PNIPAM hydrogel, and the insert is the atomic percentage of carbon, nitrogen, and oxygen at the HD side and LD side. (**c**) Transmittance vs. temperature for the LCST-type hydrogels with different monomer mass ratios (DMAEMA/NIPAM). (**d**) The optical images of hydrogel at room temperature and after heating. (**e**) Stress–strain curves of the hydrogel with different DMAEMA/NIPAM mass ratios at room temperature. (**f**) Stress–strain curves of the hydrogel with different DMAEMA/NIPAM mass ratios after heating. Stress–strain cycle curves of hydrogels with different concentrations. (**g**) DMAEAM:NIAPM = 5:1, (**h**) DMAEAM:NIAPM = 10:1, (**i**) DMAEAM:NIAPM = 15:1.

**Figure 3 polymers-14-05312-f003:**
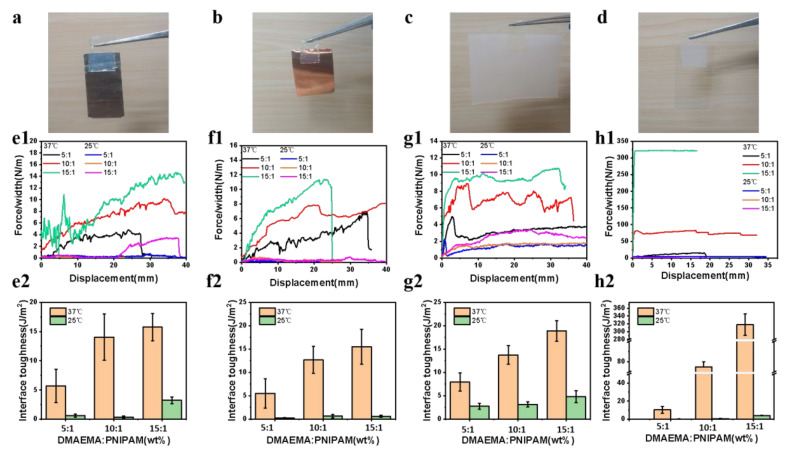
Adhesion performance of self-powered gradient hydrogel based on PDMAEAM/PNIPAM. Photos of hydrogels adhered on (**a**) platinum, (**b**) copper, (**c**) silicone rubber and (**d**) glass. (**e1**) Peeling force curves and (**e2**) interface toughness of hydrogel with different proportions at room temperature and high temperature in binding platinum; (**f1**) peeling force curves and (**f2**) interface toughness of hydrogel with different proportions at room temperature and high temperature in binding copper; (**g1**) peeling force curves and (**g2**) interface toughness of hydrogel with different proportions at room temperature and high temperature in binding silicone rubber; (**h1**) peeling force curves and (**h2**) interface toughness of hydrogel with different proportions at room temperature and high temperature in binding glass.

**Figure 4 polymers-14-05312-f004:**
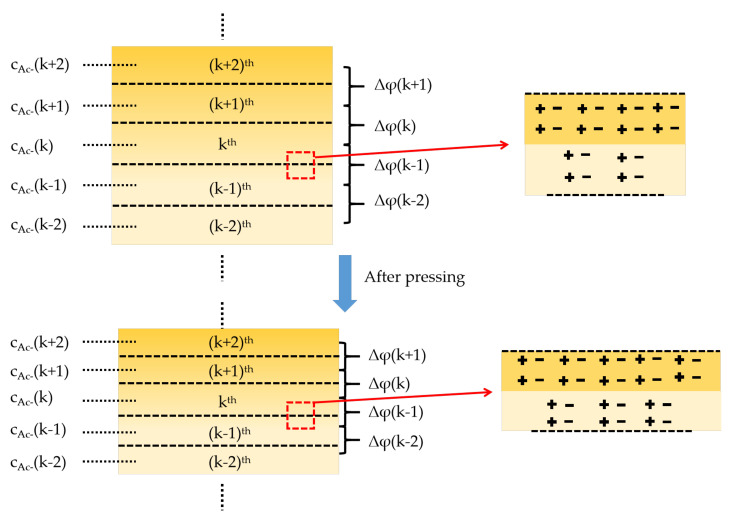
Schematic diagram of pressure and sensing mechanism of self-powered gradient hydrogel.

**Figure 5 polymers-14-05312-f005:**
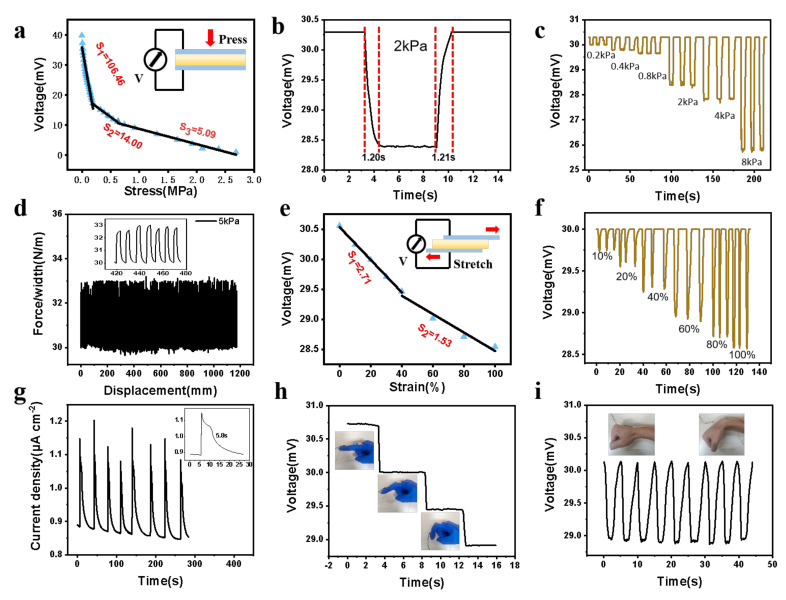
Pressure- and strain-sensing performance of the self-powered hydrogel sensors. (**a**) Output voltage of sensors as a function of pressure. (**b**) Output voltage under a 2 kPa static pressure. (**c**) Output voltage at different pressures. (**d**) Repeated loading/unloading of 5 kPa pressure for 200 cycles. (**e**) Gradually increased tensile strain and (**f**) different tensile strain. (**g**) Current signals of sensor in response to the 5 kPa pressure. Recorded voltage signals of the sensors in response to (**h**) finger joint motions and (**i**) wrist joint motions.

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
