# Peer review of "Self-Powered Gradient Hydrogel Sensor with the Temperature-Triggered Reversible Adhension"

_polymers, 2022, doi:10.3390/polym14235312_

Round 1

Reviewer 1 Report

Comments: In this manuscript, Dong sun et al described developed self-powered gradient hydrogel sensors with reversible adhension capacity based on the copolymers with phase transition temperature, which are polymerized by 2-(Dimethylamino)ethyl metharcylate (DMAEMA) and N-Isopropy-lacrylamide (NIPAM). The authors conducted some experiments and analyzed the experimental results to prove the validity and novelty of the experiments. The reviewer recommends that this manuscript be accepted after some revisions as below.

 Q1: How was the UV irradiation time given to form HD and LD?

 Q2: How similar is the fabricated film thickness to the actual human epidermal layer?

 Q3. In Figures 2e and 2f, experiments on strain recovery after removing the force should be added.

 Q4: The adhesive force with silicone will be similar to the adhesive composition with human skin. What happens to the skin pain when 18.8 J/m2 is applied?

 Q5: The mechanism of the self-powered film should be added.

 Q6: Some important references about self-powered film should be additionally cited below.

- Journal of Materials Chemistry A 10 (28), 14894-14905

- Applied Materials Today 29, 101633

Reviewer 2 Report

The obtained results are significant and interesting. So I recommend its publication in with minor revision and review as listed below

1.      Preparation of the gradient polyelectrolyte hydrogels- did you used any adhesive materials

2.      What is effect of UV light

3.      The self-powered hydrogel sensors can be easily integrated by connecting the upper  and lower sides of the hydrogel to two platinum copper foil electrodes- one side acting what type of nature

4.      Characterization- what about EIS  Study

5.      The hydrogels were prepared by free radical polymerization- how many steps involved

6.      How you justify about your novelty of the sensor ?

7.      What is the sensitivity of the sensor .

8.      How you did the pretreatment of the electrode

9.       Can please give details about repeatability

10.  Comparison table also missing

Round 2

Reviewer 1 Report

The author addressed all comments well.